# Complement Blockade Is a Promising Therapeutic Approach in a Subset of Critically Ill Adult Patients with Complement-Mediated Hemolytic Uremic Syndromes

**DOI:** 10.3390/jcm11030790

**Published:** 2022-02-01

**Authors:** Renaud Prével, Yahsou Delmas, Vivien Guillotin, Didier Gruson, Etienne Rivière

**Affiliations:** 1CHU Bordeaux, Medical Intensive Care Unit, F-33000 Bordeaux, France; vivien.guillotin@chu-bordeaux.fr (V.G.); didier.gruson@chu-bordeaux.fr (D.G.); 2University Bordeaux, Centre de Recherche Cardio-Thoracique de Bordeaux, Inserm UMR 1045, F-33000 Bordeaux, France; 3CHU Bordeaux, Nephrology Transplantation Dialysis Apheresis Unit, F-33076 Bordeaux, France; yahsou.delmas@chu-bordeaux.fr; 4CHU Bordeaux, Internal Medicine Department, F-33000 Bordeaux, France; etienne.riviere@chu-bordeaux.fr; 5University Bordeaux, Biology of Cardiovascular Diseases, InsermU1034, F-33604 Pessac, France

**Keywords:** thrombotic microangiopathies, haemolytic uremic syndrome, complement, eculizumab, critical care

## Abstract

Thrombotic microangiopathy (TMA) gathers consumptive thrombocytopenia, mechanical haemolytic anemia, and organ damage. Hemolytic uremic syndromes (HUS) are historically classified as primary or secondary to another disease once thrombotic thrombocytopenic purpura (TTP), Shiga-toxin HUS, and cobalamin C-related HUS have been ruled out. Complement genetics studies reinforced the link between complement dysregulation and primary HUS, contributing to reclassifying some pregnancy- and/or post-partum-associated HUS and to revealing complement involvement in severe and/or refractory hypertensive emergencies. By contrast, no firm evidence allows a plausible association to be drawn between complement dysregulation and Shiga-toxin HUS or other secondary HUS. Nevertheless, rare complement gene variants are prevalent in healthy individuals, thus providing an indication that an investigation into complement dysregulation should be carefully balanced and that the results should be cautiously interpreted with the help of a trained geneticist. Several authors have suggested reclassifying HUS in two entities, regardless of they are complement-mediated or not, since the use of eculizumab, an anti-C5 antibody, dramatically lowers the proportion of patients who die or suffer from end-stage renal disease within the year following diagnosis. Safety and the ideal timing of eculizumab discontinuation is currently under investigation, and the long-term consequences of HUS should be closely monitored over time once patients exit emergency departments.

## 1. Introduction

Thrombotic microangiopathy (TMA) is a group of rare but potentially life-threatening diseases that are characterized by consumptive thrombocytopenia, mechanical haemolytic anemia, and organ damage, mostly the kidneys, brain, and heart [1]. Among these diseases, primary and secondary hemolytic uremic syndromes (HUS) are evoked once thrombotic thrombocytopenic purpura (TTP) (a disintegrin and metalloprotease with thrombospondin type I repeats-13 (ADAMTS13) activity <10%), Shiga-toxin HUS (specific stool-or rectal swab culture and shigatoxin polymerase-chain reaction (PCR)), and cobalamin C-related HUS (very high homocysteinemia) are ruled out. HUS can be linked to another disease or condition (secondary HUS, about 90% patients) or not (primary HUS, incidence <1 per million population per year) (Figure 1) [2]. Studies of complement genetics have led to the first specific treatment for this condition: an anti-C5 antibody called eculizumab [3,4]. Complement genetic studies have also reinforced the link between complement dysregulation and primary HUS but have also contributed to reclassifying some pregnancy- and/or post-partum-associated HUS and to reveal complement involvement in severe and/or refractory hypertensive emergencies. By contrast, no firm evidence allows for possible associations to be drawn between complement dysregulation and Shiga-toxin HUS or other secondary HUS. Moreover, rare complement gene variants are also found in healthy individuals, meaning that caution is required when interpreting the results.

The aim of this review is to provide a pragmatic approach to intensivists about when to investigate for complement dysregulation with specialized alternative complement analysis in critically ill patients harbouring thrombotic microangiopathy (TMA) features, how to cautiously interpret these results, and to discuss when to consider anti-C5 therapy in patients with HUS who have been admitted to emergency or intensive care units.

## 2. A Role for Complement in Secondary HUS Pathophysiology?

Secondary HUS are associated with numerous conditions, such as drugs, cancer, infections, autoimmune disorders, hematopoietic stem cell transplantation (HSCT) or solid organ transplantation, pregnancy, or hypertensive emergencies. The involvement of complement regulation in HUS associated with these diseases has still not been extensively investigated but does not seem to be identical in each condition [5].

HUS that are associated with drugs, cancer, infections, auto-immune disorders, or HSCT do not seem to be caused by complement dysregulation [5] due to their very low risk of relapse once the associated disease has been treated or the associated drug has been withdrawn [6,7]. In these conditions, treatment with eculizumab has not shown conclusive efficacy and could even worsen the prognosis of treated patients in terms of dialysis requirements and neurological involvement [6,7].

In kidney transplant recipients, most cases of de novo TMA are associated with immunosuppressive drugs (calcineurin inhibition) [8] or antibody-mediated rejection [9]. Nevertheless, the risk of TMA after kidney transplantation is incredibly increased in patients who have previously experienced HUS in the native kidney compared to in other patients [10]. Moreover, rare variants in complement genes can be found in 29% of kidney transplant recipients with de novo TMA and in up to 68% with recurrent TMA [11,12]. Eculizumab is efficient in reducing the risk of TMA recurrence in kidney transplant recipients [13], thus reinforcing the hypothesis of complement dysregulation involvement in this subset of patients. In some countries, a highly individualized prophylactic complement blockade strategy has been applied in patients with HUS undergoing kidney transplantation, with dramatic improvements in terms of graft survival [14]. To the contrary, patients with TMA related to antibody-mediated rejection and donor-specific allo-antibodies in kidney transplantation have poor response to eculizumab [15].

Pregnancy has been linked to a first episode of HUS in about 20% of women [16], and rare variants in complement genes have been detected in 40–56% of patients [17,18]. Nevertheless, preeclampsia and HELLP (hemolysis, elevated liver enzymes, low platelets) syndrome are much more frequent causes of pregnancy- or post-partum-related TMA than HUS and should therefore be ruled out first [19]. Even if preeclampsia or HELLP can be a coexisting condition that triggers the development of pregnancy-associated HUS [20,21], these two conditions have been linked to complement activation but not complement dysregulation, as suggested by the fact that (i) most evidenced variants of complement genes were of uncertain or no significance [22], (ii) increased complement deposition in in vitro assays persisted more than 90 days after the clinical remission of preeclampsia, eclampsia, and HELLP syndrome [23], and (iii) these conditions can occur in patients who have been treated with eculizumab for paroxysmal nocturnal haemoglobinuria or complement mediated HUS [24,25].

The HUS diagnosis should be reassessed in women who have severe kidney involvement or no improvement in kidney function after delivery, as eculizumab seems to have similar efficacy in pregnancy-associated HUS as it does in primary HUS [17,18,26,27,28,29]. The onset of TMA features occurring after delivery (up to 3 months) is also highly suggestive of complement-mediated HUS. The ratio of soluble fms-like tyrosine kinase 1 (sFlt-1) to placental growth factor (PlGF) can be used to exclude preeclampsia (if low: excellent negative predictive value) [30]; it is helpful to discriminate HELLP from aHUS after 20 weeks of gestation. In fact, preeclampsia is due to defective trophoblastic invasion and the incomplete remodelling of the spiral arteries occurring early in pregnancy. These abnormalities have been associated with decreased levels of PlGF (the main pro-angiogenic factor) and increased levels of anti-angiogenic factors such as sFlt-1. sFlt-1 is also able to directly inhibit PlGF vasodilatory effects and angiogenesis [30]. As HUS is not due to such a defect in trophoblastic vascular invasion, there is no reason why the sFlt-1/PlGF ratio would be elevated in this condition. However, preeclampsia or HELLP can co-exist with TTP or complement-mediated HUS and cannot be excluded when an elevated ratio is found [31].

Hypertensive emergency is common in patients with HUS (about 50%) [32,33] but HUS remain a rare cause of hypertensive emergency (<3%) [34]. Common causes such as treatment cessation, primary hyperaldosteronism, oestrogen contraception, pheochromocytoma, bilateral renal artery stenosis, and glomerulopathies should first be ruled out. Clinical features that reinforce the hypothesis of hypertension linked to primary complement mediated HUS are female gender [35,36,37,38], age <45 years [34,39,40], absence of previously known hypertension and/or treatment cessation [32,35,36,41], persistent profound hematologic abnormalities despite strict control of the blood pressure, absence of left ventricular hypertrophy [35,36,40,41], requirement for renal replacement therapy [34], and neurological manifestations [42]. Of note, systemic hemolysis is frequently absent (73%) [42]. However, these descriptions are mostly retrospective and require further validation in prospective cohorts that would include patients with hypertensive emergencies and TMA. Despite the fact that kidney disease is common in patients with hypertensive emergency and is sometimes associated with hemolysis [43], a kidney biopsy is rarely performed, and HUS can be missed [42]. Performing a kidney biopsy is a safety concern in patients with uncontrolled high blood pressure, and even more so when it is associated with thrombocytopenia, but one should probably be performed when blood pressure is stabilized in cases where there are suggestive features of HUS. When performed, the absence of glomerular fibrin thrombi could help to exclude complement-mediated HUS, as it seems to be a constant feature in kidney pathology in this subset of patients. Malignant hypertension has rather arteriolar intima oedema and arterial tree lesions, whereas complement-mediated endothelial lesions concern the glomerular capillaries [37]. In fact, the treatment of a hypertensive emergency mainly relies on blood pressure control (first-line therapy with increasing doses of an oral renin–antiotensin system inhibitor should be considered [44]), but its efficacy is limited in patients with severe kidney disease. It is precisely these patients who have the highest prevalence of rare variants in complement genes, thus pointing to complement genes as a potential therapeutic target [45,46]. Results from retrospective studies assessing the efficacy of complement blockade in the setting of hypertensive emergencies are encouraging but remain to be confirmed in future prospective trials [32,33,42].

Finally, a shared finding in secondary HUS with complement involvement is that patients with abnormal complement regulation more often have poor kidney outcomes and a higher risk of recurrence [12,17,42,47]. Moreover, complement dysregulation could be seen as an underlying predisposing condition that increases the risk of HUS in case of endothelial lesion, as this occurs during those associated diseases.

## 3. A Role for Complement in Primary HUS Pathophysiology

When no associated condition is found, HUS are classified as primary. The extremely diverse and incomplete penetrance of genetic mutations in the complement genes enhance the hypothesis that abnormal complement regulation could be a predisposing risk to HUS when an unknown second hit occurs and finally leads to endothelial aggression [48,49].

In fact, about 50% of patients with primary HUS have rare germline variants or hybrid genes in alternate pathway complement genes mainly coding for factor H, factor I, membrane cofactor protein (MCP/CD46), C3, factor B [32,50,51,52,53,54], or, less commonly in adults, autoantibodies against complement factor H. Most of these rare germline variants occur in the factor H gene (25% of all mutations) [55]. Mutations in the *CFH* gene were discovered in 1999 and were the first evidence for the implication of complement dysregulation in HUS [56]. Other frequent variants are evidenced in the gene encoding MCP/CD46 (about 10%) [57,58,59] and complement factor I (5 to 10%) [60]. Less than 5% of patients have activating mutations prolonging the half-life of C3 or complement factor B [61,62]. More rarely, 5 to 10% of patients have autoantibodies against factor H. The remaining 40 to 50% of patients with primary HUS do not have any rare variants nor autoantibodies evidenced, but this does not exclude the possibility of other complement regulation abnormalities in the pathophysiology.

## 4. When to Consider Exploration of Complement Regulation in Critically Ill Patients with TMA Features?

Even if the diagnosis of HUS and a potential role for complement dysregulation should be evoked early in front of TMA features, TTP (blood ADAMTS13 activity), cobalamin C-related HUS (homocysteinemia) and Shiga-toxin HUS (specific stool or rectal swab culture and shigatoxin PCR) should be first excluded in patients presenting with TMA features.

Regarding secondary HUS, with the exception of a few cases of severe clinical presentations with the absence of response to standard care and/or recurrence, complement dysregulation should probably not be investigated in drug-, cancer-, bacterial infections-, HSCT-, and autoimmune disorder-associated HUS. As previously discussed, complement dysregulation is much more prevalent in hypertensive emergencies and pregnancy- or post-partum-associated HUS and, in case of dubious clinical features, the exploration of complement regulation could be undertaken. Nevertheless, complement activation (i) can be observed in the presence or absence of genetic dysregulation, (ii) is not necessarily harmful, and (iii) can be transient and self-remitting [55]. The exploration of complement dysregulation should probably be restricted to severe and/or non-self-remitting hypertensive emergency or pregnancy- or post-partum-associated HUS. To the contrary, the exploration of complement dysregulation should be systematically undertaken in patients with suspected primary HUS [55].

The exploration of complement regulation includes routine measures of global complement activity, more specialized measures of complement protein quantification and autoantibodies screening—which must be performed early on in samples that have been collected before any interfering intervention—and finally, genotyping screening as a second step. The measurement of global complement activity, the serum dosage of C3, C4, and CH50, is often performed first. Of note, routine measures of complement activity are often poorly reliable in HUS settings, mainly because of the underlying inflammation—which increases C3 and C4 levels—leading to the presence of a dissociated low C3 (versus C4) in only one third of patients with complement-mediated HUS [46,50,63]. This explains why more specialized investigations should be performed in a second phase through the measurement of serum soluble C5b9 serum concentrations of the H, I, B factors; the search for a CD46 expression defect on leukocytes; and screening for auto-antibodies against factor H. It is mandatory to have early samples available for these second phase analyses, before interfering treatments are started, such as plasma infusion or exchange, or extra-corporeal therapies. Indeed, these treatments could completely flaw the analyses when performed on samples that have been collected at later timepoints. Those serum and plasma samples should be stored at −80 °C to enable the retrospective functional analysis of complement regulation (e.g., C3, C4, soluble C5b-9).

In case of a high-suspicion of complement-mediated HUS, genetic sequencing will be performed, but as a second step because it is mostly useful to determine when to stop complement blockade therapy and because these tests are not impaired by any undergoing treatment. These are the sequencing of the *CFH*, *CFI*, *CD46*, *CFB, and C3* genes and the detection of hybrid genes and/or the loss of *CFHR1* and *CFHR3* caused by non-allelic homologous recombination [5,50,51,52,64]. In case of the homozygous deletion of *CFHR1* and *CFHR3*, serum factor H reactivity should be tested [65] (Figure 2).

The main message is that the genetic sequencing results must be cautiously interpreted in collaboration with a trained geneticist who is aware of complement biology. In fact, several limitations should be considered. First, HUS are not monogenic diseases, as familial forms have only been reported in less than 20% of patients [50,54], and some carriers of variants associated with monogenic diseases will never develop HUS because of an incomplete penetrance [67]. Second, more than 600 variants have been identified in complement genes in patients with primary HUS [53]. Third, data from whole-genome sequencing projects indicate that rare variants in the five complement genes with minor allele frequencies (MAFs) of <1% and <0.1% are present in, respectively, 12 and 3.7% of healthy individuals; thus, the frequency of the allele cannot necessarily help to assess this causative role. Fourth, and importantly, the analysis of functional alterations—either a quantitative defect (level of protein expression) or qualitative defect (impaired activity of the encoded protein due to the variant)—should be taken into consideration. However, the assessment of the qualitative defect relies more on predictive data than on actual proof of pathogenicity [68,69]. Based on those considerations, complement gene variants have been classified along a gradient from those with an almost certain pathogenic role to those that are very likely to be benign [22], explaining why discussion with a trained geneticist who is aware of complement biology is mandatory when interpreting these results.

A functional assessment of ex vivo complement activation could be a promising method in years to come to detect the unrestrained complement activation on the endothelium and thus complement-mediated HUS, irrespective of rare variants in complement genes [46,63]. These tests assess the induction by the plasma or serum from patients with TMA of complement deposition (mostly C5b-9) on endothelial cells in vitro [23,46,63]. Nevertheless, these assays currently have high inter-laboratory and even intra-laboratory variability, possibly because of the variable activation states of the endothelial cells, therefore requiring standardization before their translation into clinical practice [23,70].

## 5. What Is the Impact of Complement Dysregulation in HUS Patients Care?

Supportive care remains the standard of care of critically ill patients with HUS, including watchful monitoring, maintenance of the hydro-electrolytic balance and, in case of severe kidney injury, renal replacement therapy. Due to progress in the critical care of such patients, neurological complications and multi-organ failures due to HUS are now rare but still can exist and may still require mechanical ventilation and vasopressive support. Over the last few decades, few advances have been made in terms of specific treatments of HUS until the identification of complement dysregulation as a cornerstone of pathophysiology in primary HUS and in some secondary ones, as discussed above.

The terminal blockade of the complement pathway by eculizumab or ravulizumab has considerably improved the prognosis of patients with primary HUS. It also decreased the proportion of patients who die or suffer from end-stage renal disease within 1 year of diagnosis from 56% to less than 15% [4,71,72]. As a consequence, the quick recognition of patients with potential complement dysregulation has become of crucial importance among patients with TMA. Patients with early initiation of the complement blockade are given the best possible chance to recover from kidney injury [73].

If complement blockade is now extensively used in primary HUS, the place of complement blockade in specific subtypes of secondary HUS remains to be fully understood [6]. In case of hypertensive emergencies, the persistence of severe conditions and/or hematological abnormalities, despite adequate blood pressure control and the elimination of other more common causes, or a familial history of a HUS can motivate the early initiation of anti-complement drugs. However, the timing and place of this strategy in pregnancy and post-partum-related HUS remains to be investigated.

When complement blockade is adequately initiated, clinical improvement is usually observed within days or weeks following its initiation, with the best patient response being achieved within 3 to 6 months. New anti-C5 agents are currently under investigation, including agents that can be administered sub-cutaneously that could improve patient quality of life during maintenance therapy. In case of an incomplete or lack of response to documented complement blockade (CH50, dosage of free eculizumab), alternative diagnoses should urgently be investigated, including hyperhomocysteinemia with low methioninemia and methylmalonic aciduria (due to recessive variants in methylmalonic aciduria and homocystinuria type C gene (MMACHC)) [74] it this has not already been considered (Figure 1).

Due to its blockade of the terminal complement mechanisms, eculizumab increases the risk of infections. Overall serious infection was observed in 8% of patients in a ten-year pharmacovigilance analysis [75]. There is also a notable risk for pyogenic and encapsulated bacteria, in particular *Neisseria meningitidis,* with a 0.25/100 patient-year rate [75]. Immunization against the common serotypes (ABCWY) is thus recommended for patients or their close relatives, along with individualized chemoprophylaxis. Little is known about prevention against other infections, but the anti-pneumococcal vaccine should probably be provided, even if invasive infections with encapsulated bacteria such as *Streptococcus pneumoniae* or *Haemophilus influenza* remain rare [76], possibly thanks to the effect of opsonisation [77].

In the ICU setting, vaccination often occurs at the same time as treatment initiation, and data about the quality of response to vaccines in patients treated with eculizumab and in a critically condition are scarce. Response to vaccination should probably be monitored after ICU discharge, and booster doses should be considered; however, no serological assay to analyse the quality of vaccination against *N. meningitidis* is available so far in daily practice. Chemoprophylaxis with oral penicillin V or ciprofloxacin/azithromycin until at least 4 weeks after immunization is usually prescribed [78,79]. Most of the time, chemoprophylaxis is given until 2 months after eculizumab discontinuation or 8 months after ravulizumab discontinuation (end of terminal complement blockade activity), as no data exist regarding an eventual individualization according to antibodies titres or the patient’s immunosuppressive state. In addition, the optimal level of serum titres is not precisely known, as individuals with late complement-deficiency require higher antibodies titres for opsonophagocytosis [80]. Further insights are needed before any recommendation can be made at this point. However, every physician dealing with these patients must check that they are adequately protected against *Neisseria* sp. infection through adequate vaccination and chemoprophylaxis.

Sexual life is barely taken into account in ICU, but sexually active patients and their partners should also receive counselling and screening for gonococcal disease before discharge from hospital. Indeed, this species of *Neisseria* can also cause systemic infection or septic shock in patients treated with complement-blocking agents [81,82].

In case of acute infection of any cause, and whenever possible, the complement-blocking agent should be suspended, and infection should be reported to regulatory safety agencies.

Because of the infectious risk and the elevated costs of treatment, the question of complement blockade discontinuation has rapidly been raised. Some small retrospective series have suggested that treatment discontinuation could be safe and feasible, even if relapses seem to occur in patients carrying rare complement gene variants [83,84,85]. A recent prospective study assessed eculizumab discontinuation in 55 responding patients with HUS after at least 6 months of disease control [86]. During a mean follow-up of 18 months, 13/55 (23%) of patients relapsed, mainly patients with rare germline variants (about 50% of these patients), whereas the risk was very low (<5%) in patients without those rare germline variants. Female sex was also associated with an increased risk of relapse in multivariate analysis. Elevated soluble C5b9 > 300 ng/mL at the time of complement blockade discontinuation was also associated with the risk of relapse. Even if this dosage is not always available in routine, it is of great interest in clinical practice to assess the complement activation level. Among the 13 patients who relapsed, 11 recovered baseline renal function, thus allowing the complement blockade to be discontinued even in patients with rare germline variants even if those patients should be closely monitored for potential relapse.

HUS are associated with an increased long-term risk of stroke and myocardial infarction. Interestingly, another subset of TMA patients with TTP also seems to have high long-term cardio-vascular risk as well [87,88,89]. This cardiovascular risk could be increased in patients with subsequent sequelae of acute kidney injury. It remains unclear whether TMA could accelerate endothelial aging or if patients with endothelial deterioration due to progressive atherosclerosis are at a higher risk of TMA. The impact of the thorough control of cardiovascular risk factors in HUS and TTP survivors has not been assessed yet despite this pathophysiological rationale, nor has been the impact of complement blockade on increased cardiovascular risk. However, these measures are quite simple to implement during patient follow up and should not be forgotten.

## 6. Conclusions

Complement genetics studies have reinforced the link between complement dysregulation and primary HUS as well as some secondary HUS, such as hypertensive emergencies and pregnancy- or post-partum-related HUS, contributing to the reclassification of these entities as complement-mediated HUS. The use of eculizumab dramatically lowered the proportion of patients who die or suffer from end-stage renal disease within 1 year of diagnosis in complement-mediated HUS. First round of laboratory analysis must be carried out in TMA critically ill patients, including routine complement analysis. For patients with primary HUS or in subsets of patients in pregnancy/post-partum or hypertensive emergency settings, early C5 blockade can be considered. Secondary steps will address genetic analysis for these patients to help personalize the duration of these therapies, even if the safety and ideal timing of complement-blocking agent discontinuation are currently under investigation. Finally, the long-term consequences of HUS should not be underestimated, and a close follow-up should be planned as soon as patients are discharged from these emergency units.

## Figures and Tables

**Figure 1 jcm-11-00790-f001:**
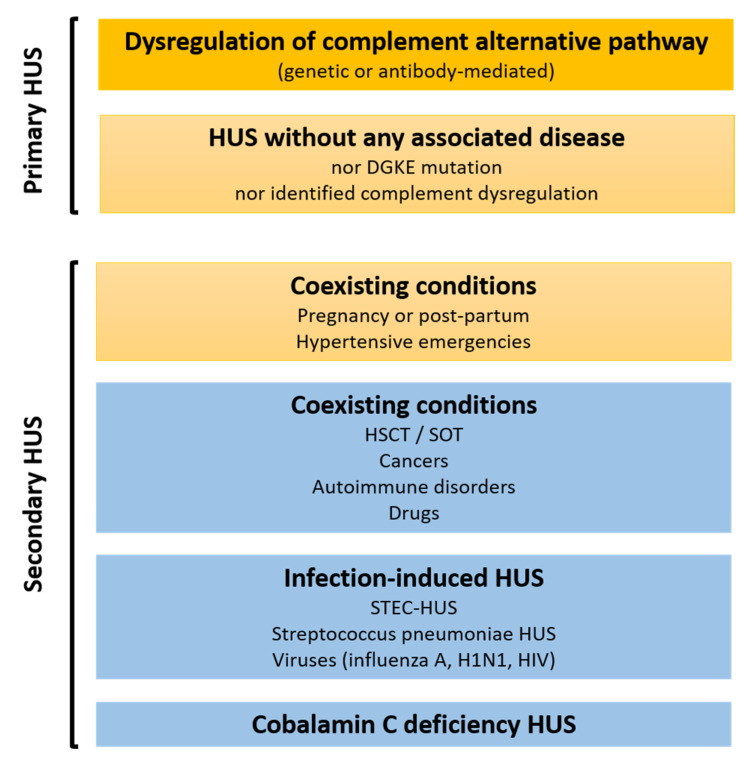
Classification used to differentiate primary and secondary hemolytic uremic syndromes according to the probability of complement-mediation. Orange: demonstrated or highly plausible complement-mediation. Pastel orange: complement-mediation highly plausible or demonstrated in a subset of patients. Blue: no evidence of complement-mediation in HUS. Abbreviations: DGKE, diacylglycerol kinase ε; HSCT: hematopoietic stem cell transplantation; HUS, hemolytic uremic syndrome; SOT, solid organ transplantation; STEC HUS, shiga-toxin *Escherichia coli* hemolytic uremic syndrome.

**Figure 2 jcm-11-00790-f002:**
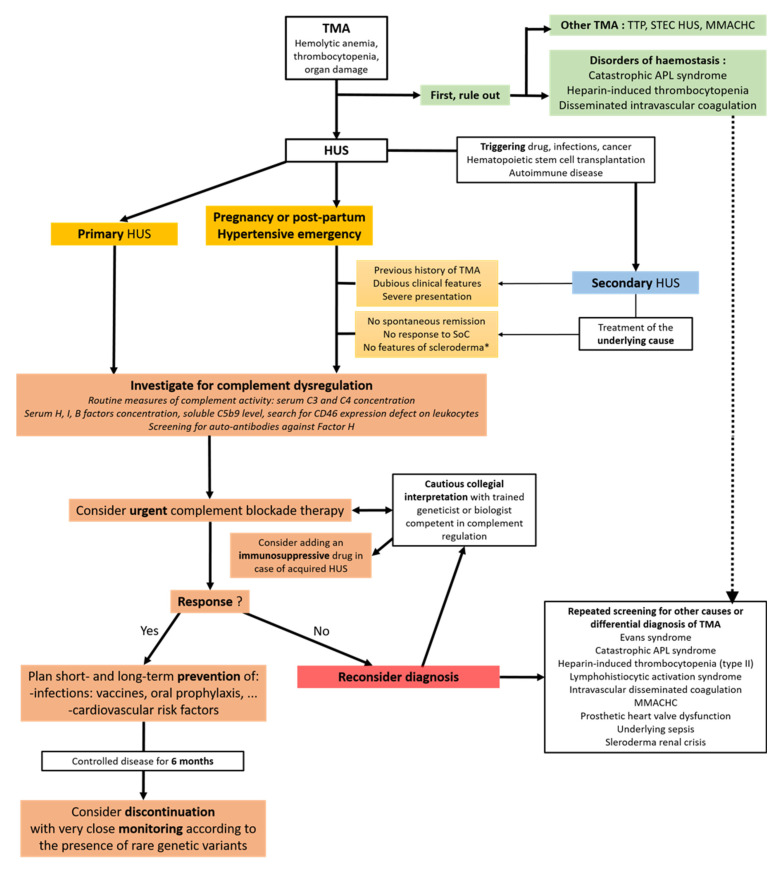
Proposed algorithm to cautiously explore and treat complement dysregulation in patients with TMA features presenting in emergency or intensive care units with a suspicion of atypical HUS. * Clinical and biological features including anti RNA polymerase 3 antibodies. Abbreviations: TMA, thrombotic microangiopathy; TTP, thrombotic thrombocytopenic purpura; STEC HUS, shiga-toxin *Escherichia coli* hemolytic uremic syndrome; MMACHC, cobalamin C-related HUS; HUS, hemolytic uremic syndrome; SoC, standard of care; APL, anti-phospholipid. Inspired from Timmermans et al., *J. Clin. Med.*, 2021 [5], Fakhouri et al., *Nat. Rev. Nephrol.*, 2021 [55] and Nester et al., *Mol. Immunol.*, 2015 [66].

## Data Availability

Not applicable.

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
