# Peer review of "Complement Blockade Is a Promising Therapeutic Approach in a Subset of Critically Ill Adult Patients with Complement-Mediated Hemolytic Uremic Syndromes"

_jcm, 2022, doi:10.3390/jcm11030790_

Round 1
Reviewer 1 Report
The manuscript is well written and covers an important field in clinical medicine. I have only a few comments:
- It should be mentioned, that storage of serum and plasma (-80°C) from the patient at the initial presentation is mandatory in order to enable retrospective functional analysis of complement regulation (e.g. C3,C4, Sc5b-9). Due to transfusions or plasma-treatment the analysis may be misleading when done afterwards.
- Please correct the following:
Page 3, line 113 and 117: use consistent terminology for "hemolysis"
Page 3 line 111 and page 6 line 250: use consistent terminology for "hematological"
Reviewer 2 Report
In this review, the authors discussed the classification of HUS according to the relation with complement system and treatment of complement related HUS. Interest to the role of complement in non-primary HUS is increased in last years. Eculizumab is a lifesaving drug for patients with primary-complement related HUS. By contrast, evidence on using eculizumab for secondary HUS cases is limited.
Overall the paper is well written and discussed.
Some corrections are required.
Major Corrections;
- I recommend to use ‘Hemolytic uremic syndrome’ instead of ‘Hemolytic and Uremic Syndromes’
- In this paper, the classification of HUS was done based on the literature; ‘Sheerin, N.S.; Kavanagh, D.; Goodship, T.H.J.; Johnson, S. A National Specialized Service in England for Atypical Haemolytic 349 Uraemic Syndrome-the First Year’s Experience. QJM 2016, 109, 27–33’. This reference does not include aforementioned information.
‘George JN, Nester CM. Syndromes of thrombotic micro- angiopathy. N Engl J Med. 2014;371:1847–1848’ fits better for the classification of HUS (primary/secondary cases).
The classification of HUS is not clear. Please give in detail about the primary/secondary cases. Also, a figure demonstrating the classification will be good for better understanding.
- Page 3, line 99; Please give in detail the role of sFLT-1 and PIGF in pregnancy. Why the ratio of sFLT-1 to PIGF important in preeclampsia?? Is there any relation with HUS?
Minor;
- Introduction Page 1 line 40, for TTP and cobalamin C related HUS differential diagnosis criteria were given. Please write an explanation for how to exclude ‘Shiga-toxin HUS’
Round 2
Reviewer 2 Report
No comment